# Pentraxin-3 in the Spinal Dorsal Horn Upregulates Nectin-1 Expression in Neuropathic Pain after Spinal Nerve Damage in Male Mice

**DOI:** 10.3390/brainsci12050648

**Published:** 2022-05-15

**Authors:** Min Zhu, Hongli Yu, Ying Sun, Wenli Yu

**Affiliations:** Department of Anesthesiology, Tianjin First Central Hospital, Tianjin 300192, China; zhu.min1987@163.com (M.Z.); yuhongli11262022@163.com (H.Y.); sunying2771@163.com (Y.S.)

**Keywords:** nectin-1, neuropathic pain, pentraxin-3, spinal nerve damage

## Abstract

**Purpose:** Neuropathic pain often originates from nerve injury or diseases of the somatosensory nervous system. However, its specific pathogenesis remains unclear. The requirement for excitatory synaptic plasticity in pain-related syndromes has been demonstrated. A recent study reported that pentraxin-3 is important in glutamatergic synaptic formation and function. Meanwhile, nectin-1 mediates synaptogenesis in neurological disorders. The present study aimed to evaluate whether pentraxin-3 and nectin-1 modulate spinal nerve damage-related neuropathic pain in male mice. **Methods:** L_4_ spinal nerve ligation (SNL) in male mice was performed to induce experimental neuropathic pain. Mechanical allodynia and heat hyperalgesia following SNL were based on paw withdrawal (PW) threshold and PW latency, respectively. Spinal pentraxin-3 levels and nectin-1 expression following SNL were examined. Pentraxin-3 and nectin-1 knockdown models were established by the shRNA method. These models were used with a recombinant pentraxin-3 cell model to investigate the underlying mechanisms of SNL. **Results:** The SNL operation generated persistent decreases in mechanical PW threshold and thermal PW latency, with subsequent long-lasting elevations in spinal pentraxin-3 and nectin-1 expression levels. Pentraxin-3 knockdown reduced SNL-associated neuropathic pain behaviors as well as nectin-1 amounts in the spinal dorsal horn. Nectin-1 deficiency impaired mechanical allodynia and thermal hyperalgesia following spinal nerve injury. The application of recombinant pentraxin-3 in the spinal cord triggered an acute nociception phenotype and induced spinal overexpression of nectin-1. The intrathecal knockdown of nectin-1 prevented exogenous pentraxin-3-evoked pain hypersensitivity. **Conclusions:** The findings suggest spinal pentraxin-3 is required for SNL-triggered neuropathic pain via nectin-1 upregulation in male mice.

## 1. Introduction

Neuropathic pain, which features tactile allodynia, heat hyperalgesia and spontaneously occurring burning pain, remains a debilitating pain state and is refractory to commonly available therapies [1,2]. To date, multiple models of long-lasting pain in animals have been generated for investigating the pathogenesis of neuropathic pain. However, the etiological mechanisms of neuropathic pain encompass neuroinflammation, neuronal plasticity and synaptic formation and are poorly elucidated [3,4,5].

The α-amino-3-hydroxy-5-methyl-4-isoxazolepropionic acid (AMPA) receptor is considered one of the most critical contributors in functional synaptogenesis’ underlying pain-related behaviors [6,7,8]. For example, the necessity of GluA1-containing AMPA receptor recruitment to postsynaptic locations has been reported to control neuroinflammation and glutamatergic neurotransmission during the process of postoperative pain following tibial bone fracture treated by orthopedic operation [9]. Pharmacological inhibition of GluA1-containing AMPA receptor insertion into synapses has been shown to reverse the remifentanil-induced decrease in paw withdrawal (PW) mechanical threshold and PW thermal latency [10,11]. However, the mechanism by which the functional hyperactivity of the AMPA receptor contributes to these processes has not been appropriately described.

Pentraxins are soluble and multifunctional pattern recognition molecules that modulate innate immunity [12]. Among different pentraxins, pentraxin-3 is particularly important for its specific production and distribution in the central nervous system and induction by pro-inflammatory pathways in several neurological diseases [13,14,15]. Of note, pentraxin-3 has been shown to promote the formation and development of functional synapses by increasing AMPA receptor trafficking in the rodent hippocampus and neuronal synapses [16]. Furthermore, clinical evidence has recently suggested that plasmic pentraxin-3 may represent a strong innovative biomarker of acute chest pain and diabetic polyneuropathy-associated pain [17,18]. Nonetheless, it is virtually unknown whether spinal pentraxin-3 is required for neuropathic pain initiation and sustenance. Nectin-1, which belongs to immunoglobulin-like cell adhesion molecules, is localized in excitatory glutamatergic synapses and constitutes an important determinant of synapse generation and function [19]. Moreover, spinal nectin-1 knockdown reduces the mechanical allodynia caused by chronic sciatic nerve constriction in a rat model [20]. However, the detailed alterations of nectin-1 in the pathophysiology of peripheral nerve trauma-related neuropathic allodynia and hyper-nociception are elusive.

In the present study, pentraxin-3 and nectin-1 changes were examined after the establishment of chronic neuropathic pain associated with spinal nerve ligation (SNL) in male mice. The effects of pentraxin-3 and nectin-1 on the development of SNL-related neuropathic pain were investigated, and the potential link between pentraxin-3 and nectin-1 was characterized with reference to this pathological condition. The present study highlighted that the pentraxin-3/nectin-1 pathway could be used for the authentication of innovative therapeutic targets for neuropathic pain.

## 2. Materials and Methods

### 2.1. Mice

Eight-to-twelve-week-old male C57BL/6 mice were provided by the Laboratory Animal Center of the Military Medical Science Academy of the Chinese People’s Liberation Army. Animal housing was kept at 23 ± 2 °C under a 12 h/12 h light-dark cycle with food and water at will. Assays involving animals had approval from the National Institutes of Health Guide for Care and Use of Laboratory Animals, with experimental protocols approved by the Institutional Animal Care and Use Committee of Tianjin First Central Hospital.

### 2.2. Surgical Procedure

Experimental SNL in mice was established as reported in a previous study [21]. In short, after the separation of left paravertebral muscles, the left L_4_ spinal nerve was exposed in sevoflurane-anesthetized animals. The left L_4_ spinal nerve underwent tight ligation with 6-0 braided silk thread, followed by stitching of the wound with 3-0 silk. A sham operation was carried out by exposing the left L_4_ spinal sciatic nerve as described above but with no ligation under sevoflurane anesthesia.

### 2.3. Reagents and Administration

Recombinant pentraxin-3 (Abcam, Cambridge, UK) was dissolved in 0.9% normal saline (vehicle) and administration was conducted via intrathecal injection. The dose was selected based on our rudimentary experiments and the recommendations from the manufacturer. The previously characterized pentraxin-3 shRNA and nectin-1 shRNA [22,23] sequences were packaged into lentiviruses (GeneChem, Shanghai, China). The negative control lentivirus and lentiviruses harboring pentraxin-3 and nectin-1 shRNAs, respectively (10 μL; 3 × 10^8^ transducing units [TU]/mL), were administered by intrathecal injection. Under brief sevoflurane anesthesia (induction at 3.0% and maintenance at 1.5%), injection through intrathecal routes was carried out at the L_4_ and L_5_ levels with 30 G needles [24].

### 2.4. Behavioral Tests

All male animals were adapted to the test conditions for 3 days prior to any nociceptive behavior examinations. To assess mechanical sensitivity, the mice were placed in a Plexiglas box and allowed 2 h for further acclimation prior to the study. The plantar surface of the left hind paw underwent stimulation with perpendicularly presented von Frey hairs, with exponentially increasing stiffness from 0.02 g to 2.56 g (Stoelting). Then, 50% of the PW’s mechanical threshold was determined by the Dixon’s up-and-down method [25]. For the heat hyperalgesia assessment, the PW’s thermal latency was assessed with a YLS-6B hot plate (Huaibei Zhenghua, Biological Instrument Equipment, Huaibei, China) [25]. Behavioral assays were performed in a blinded manner.

### 2.5. Immunoblot

All the animals were sacrificed under deep sevoflurane (3%) anesthesia. The spinal dorsal horn segments (L_3_–L_5_), amygdala and cortex were removed quickly and snap-frozen in liquid nitrogen. The tissues underwent mechanical homogenization in chilled RIPA buffer with PMSF (Abcam, Cambridge, UK). The lysates were cleared by centrifugation, and supernatants were obtained for total protein extraction. Protein amounts were assessed by the bicinchoninic acid assay. Protein separation was carried out by 10% SDS-PAGE. After transfer onto nitrocellulose membranes, the specimens underwent successive incubations with rabbit monoclonal antibodies targeting pentraxin-3 and rabbit polyclonal antibodies nectin-1 (1:1000; Santa Cruz, CA, USA), respectively, and horseradish peroxidase-linked secondary antibodies (1:2000, Jackson ImmunoResearch, PA, USA). Enhanced chemiluminescence (ECL) detection reagents (Thermo Scientific, IL, USA) were utilized for development, followed by quantification with Gene Tools Match software (Syngene, Cambridge, UK). Antibodies targeting β-actin (1:5000; Sigma, MO, USA) were utilized for normalization.

### 2.6. Quantitative Real-Time PCR (qRT-PCR)

Pentraxin-3 and nectin-1 gene expression levels in spinal cord dorsal horn, amygdala and cortex specimens were assessed. The total mRNA was extracted, as described by the manufacturer of the RNA 4 Aqueous kit (Ambion, Austin, TX, USA). The Retroscript kit (Ambion) was employed for reverse transcription of a total of 1 mg total mRNA. qRT-PCR was carried out on an API Prism 7900HT Sequence Detection system with the SYBR Green PCR Master Mix kit (Applied Biosystems, Foster city, CA, USA), as directed by the manufacturer. Amplification was performed at 50 °C (2 min) and 95 °C (10 min), followed by 40 cycles at 95 °C (15 s) and 60 °C (60 s). Glyceraldehyde 3-phosphate dehydrogenase (GAPDH) was utilized for normalization in the 2^−∆∆Ct^ method [26]. Primers were: pentraxin-3, sense 5′-CCTGCTTTGTGCTCTCTGGT-3′ and antisense 5′-TCTCCAGCATGATGAACAGC-3′; nectin-1, sense 5′-CCGTAAAGGTCAAGGGCAGAG-3′ and antisense 5′-GTGCCTGTCCCTTGTCCA-3′; GAPDH, sense 5′-AACAGCAACTCCCACTCTTC-3′ and antisense 5′-CCTCTCTTGCTCAGTGTCCT-3′.

### 2.7. Statistical Analysis

Data (mean ± SEM) were analyzed with SPSS 19.0 (SPSS, Chicago, IL, USA). Sample size estimation was based on previous reports for comparable behavioral and molecular assays [9,10]. Behavioral data analysis was carried out by one- and repeated measures two-way ANOVA, respectively, with a post hoc Bonferroni test. Biochemical data analysis was carried out by a one-way ANOVA with a post hoc Bonferroni test. *p* < 0.05 indicated significant differences.

## 3. Results

### 3.1. Initiation and Persistence of Mechanical Allodynia and Thermal Hyperalgesia Following Spinal Nerve Trauma in Male Mice

The initial experiments demonstrated no marked differences in baseline paw withdrawal mechanical threshold (*p* > 0.05, n = 6, one-way ANOVA; Figure 1A) and paw withdrawal thermal latency (*p* > 0.05, n = 6, one-way ANOVA; Figure 1B) among all three groups. The von Frey and hot plate tests detected no specific alterations in mechanical and thermal sensitivities following sham operations in comparison with baseline values. Subsequently, peripheral mechanical and heat hypersensitivities were evaluated in the mice undergoing SNL. SNL exposure generated a quick (<3 days) and prolonged (>21 days) mechanical allodynia compared with the sham animals. This was manifested by prolonged paw withdrawal threshold decrease (*p* < 0.05, n = 6, two-way ANOVA; Figure 1A) following the SNL operation. In parallel, the hot plate test demonstrated thermal hyperalgesia (>21 days, the last examination day) by a robust decrease in paw withdrawal latency following SNL surgery compared with the sham operation (*p* < 0.05, n = 6, two-way ANOVA; Figure 1B). Taken together, the behavioral data demonstrated that peripheral nerve injury caused by SNL intervention produced and sustained chronic neuropathic pain in the male mice.

### 3.2. Pentraxin-3 and Nectin-1 Are Upregulated in the Spinal Cord Dorsal Horn upon Spinal Nerve Injury in Male Mice

Neural plasticity-associated molecular variation in the spinal dorsal horn represents a pivotal contributor to the pathophysiology of peripheral nerve trauma-related pro-nociception sensation [3,5,7]. Western blot and qRT-PCR analyses revealed a rapid (within 3 days) and prolonged (>21 days) elevation in the protein and gene expression levels of pentraxin-3 following SNL in the mice (*p* < 0.05, n = 6, one-way ANOVA; Figure 2A–C), mimicking neuropathic pain behaviors. Simultaneously, spinal nectin-1 mRNA and protein amounts were overtly increased at 3 days, peaked at 7 days and retained these levels until day 21 (study end) following SNL treatment (*p* < 0.05, n = 6, one-way ANOVA; Figure 2B,D,E). These biochemical data suggested that peripheral nerve injury induced by the SNL operation caused the spinal overexpression of pentraxin-3 and nectin-1.

### 3.3. Pentraxin-3 Knockdown Reduces Neuropathic Pain Behaviors and Spinal Nectin-1 Expression after Peripheral Nerve Damage in Male Mice

To examine whether pentraxin-3 is important in neuropathic pain generation following SNL surgery, lentivirus-containing pentraxin-3-shRNA was employed for pentraxin-3 silencing in the spinal dorsal horn. Western blot and qRT-PCR analyses demonstrated reduced amounts of spinal pentraxin-3 protein and mRNA at 2 weeks following intrathecal pentraxin-3-shRNA application compared with the male mice receiving the injection of the normal saline (*p* < 0.05, n = 6, one-way ANOVA; Figure 3A–C). No changes in spinal pentraxin-3 levels were detected in the control animals following transfection with scrambled shRNA sequences (Figure 3A–C). Following 2 weeks of successful stable silencing, the mice underwent sham surgery or SNL exposure. The animals administered gene knockdown showed no significant defects in mechanical or heat sensitivities (Figure 3D,E). Similarly, the administration of scramble shRNA did not compromise SNL-related pain phenotypes. However, mechanical allodynia (*p* < 0.05, n = 6, two-way ANOVA; Figure 3D), as well as thermal hyperalgesia (*p* < 0.05, n = 6, two-way ANOVA; Figure 3E), caused by SNL treatment were successfully suppressed by spinal deficiency of pentraxin-3. Robust anti-nociception was observed from day 3 following SNL and continued for over 14 days. Pre-treatment with pentraxin-3-shRNA inhibited spinal expression of nectin-1 (*p* < 0.05, n = 6, one-way ANOVA; Figure 4A–C) on day 7 following SNL. Nerve injury drives persistent plastic alterations along somatosensory circuits from peripheral sensory terminals to the spinal dorsal horn, amygdala and cortex, causing chronic pain perception [27]. Viruses could modulate pain behaviors at the supraspinal level due to diffusion from the injection site. To rule out this possibility, the expression of pentraxin-3 in the amygdala and cortex after pentraxin-3-shRNA injection was evaluated. No difference in pentraxin-3 levels was noted between the shRNA and normal saline groups (*p* > 0.05, n = 6, one-way ANOVA; Figure 5A–C). Taken together, these detailed findings indicated that the downregulation of spinal nectin-1 was involved in the analgesic effect of pentraxin-3 knockdown.

### 3.4. Spinal Nectin-1 Deficiency Impairs the Development of Neuropathic Pain after Peripheral Nerve Damage in Male Mice

To further determine nectin-1′s potential effect on neuropathic pain behavior following SNL, a lentivirus harboring nectin-1-shRNA was utilized. The biochemical tests demonstrated reductions in spinal nectin-1 protein and gene expression levels at 2 weeks following central (intrathecal) nectin-1-shRNA administration compared with normal saline-treated mice. This effect was not noted in the control shRNA group (*p* < 0.05, n = 6, one-way ANOVA; Figure 6A–C). Following 2 weeks of successful silencing, all the mice underwent sham operations or SNL. The current behavioral data indicated that nectin-1-shRNA following intrathecal application reduced mechanical allodynia and heat hyperalgesia, with sudden increases in PW threshold (*p* < 0.05, n = 6, two-way ANOVA; Figure 6D) as well as PW latency (*p* < 0.05, n = 6, two-way ANOVA; Figure 6E) in the SNL animals. By comparison, nectin-1-shRNA did not impair normal nociceptive sensitivity (Figure 6D,E). In agreement, the intrathecal nectin-1-shRNA injection did not alter nectin-1 expression in the amygdala and cortex (*p* > 0.05, n = 6, one-way ANOVA; Figure 7A–C). The above findings suggest a requirement for spinal nectin-1 expression in the generation and persistence of neuropathic pain caused by peripheral nerve trauma.

### 3.5. Inhibition of Recombinant Pentraxin-3 Evokes Acute Pain Behaviors by Spinal Nectin-1 Knockdown in Male Mice

Subsequently, the potential contribution of exogenous pentraxin-3 in pain sensation was investigated. It is interesting to note that recombinant pentraxin-3 (10 and 100 ng but not 1 ng) caused a persistent induction of mechanical allodynia and heat hyperalgesia in the naive animals from 1 h to 12 h post-spinal application. This effect was concentration-dependent (*p* < 0.05, n = 6, two-way ANOVA; Figure 8A,B). In addition, the intrathecal injection of recombinant pentraxin-3 (100 ng) induced nectin-1 protein and gene expression in the spinal dorsal horn (*p* < 0.05, n = 6, one-way ANOVA; Figure 8C–E). Moreover, this transient pain behavior was caused by acute treatment with recombinant pentraxin-3 (100 ng) and was effectively altered following spinal nectin-1 silencing (*p* < 0.05, n = 6, two-way ANOVA; Figure 8A,B). Collectively, this detailed discovery uncovered the pivotal interaction between pentraxin-3 and nectin-1 in spinal nociception sensitization.

## 4. Discussion

The present study reported the following main findings: (i) peripheral nerve trauma elicited chronic neuropathic pain behaviors and upregulated pentraxin-3 and nectin-1 in the spinal dorsal horn of the male mice; (ii) spinal pentraxin-3 knockdown reduced peripheral nerve damage-associated neuropathic pain and nectin-1 levels in the dorsal horn; (iii) nectin-1 deficiency inhibited mechanical allodynia and heat hyperalgesia following exposure to SNL in the male animals; (iv) intrathecal administration of recombinant pentraxin-3 promoted acute pain phenotypes and spinal nectin-1 accumulation, which were impaired by central nectin-1 knockdown. The above findings recapitulate the particular value of spinal pentraxin-3/nectin-1 cascade in persistent neuropathic pain following peripheral nerve damage in male mice.

The implication of neuronal plasticity in spinal dorsal horn in nociception maintenance and its associated conditions have been previously reported [6]. Excitatory glutaminergic receptors, including AMPA and N-methyl-D-aspartate (NMDA) receptors, are important determinants in the process of pro-nociception transduction [9,28]. Recent studies have indicated that intraoperative remifentanil infusion aggravates post-surgical hyperalgesia by enhancing the activity of GluA1-containing AMPA receptors in neurons of the spinal dorsal horn [10,11]. It was shown that neuropathic pain following spinal injury is also associated with GluA1-containing AMPA receptor amounts on post-synaptic surfaces [7]. Pentraxin-3 controls the AMPA receptor recruitment at synapses as well as the induction of synaptogenesis [16]. Given the central role of the AMPA receptor in pain sensation, the present study investigated whether pentraxin-3 mediates neuropathic pain development in male mice.

This was the first study wherein SNL operation-associated peripheral nerve damage elicited the dramatic and time-oriented overexpression of pentraxin-3 in the spinal dorsal horn of male animals, which could be compliant with mechanical allodynia phenotypes. It was essential to figure out that spinal pentraxin-3 knockdown by shRNA injection effectively reduced SNL-associated mechanical allodynia and heat hyperalgesia, indicating that the pentraxin-3 cascade is required in neuropathic pain. Moreover, it was concluded that targeting pentraxin-3 may be a novel approach for controlling pathologic pain symptoms. However, the exact mechanism by which pentraxin-3 participates in neuropathic pain following SNL surgery requires further clarification.

Nectin-1 is highly involved in the formation and transmission of excitatory glutaminergic synapses. These structures are important components of the neurobiological nature of several diseases [19,23]. The prevention of sciatic nerve injury-induced pro-nociception by nectin-1 inhibition was detected in rodents [20]. This study provided multiple data for supporting spinal nectin-1, which is the downstream target of pentraxin-3, as one of the most critical factors contributing to the mechanism of SNL-associated neuropathic pain. Initially, SNL intervention persistently elicited the up-modulation of nectin-1 in the dorsal horn. Secondly, the knockdown of spinal nectin-1 successfully reduced postoperative pain following SNL surgery. Thirdly, following the inhibition of central pentraxin-3, the reversal of the effects associated with nectin-1 overexpression following spinal nerve injury was achieved. Fourthly, exogenous pentraxin-3 after intrathecal administration induced transient pain and spinal nectin-1 overload, which was reversed by the co-application of nectin-1 shRNA. These results elucidated for the first time that spinal pentraxin-3 increases the expression levels of nectin-1, triggering SNL-related pro-nociception perception in male animals. The above effects demonstrated that the pharmacological inhibition of the nectin-1 cascade could be used for developing potential therapeutic strategies. However, certain concerns are raised regarding the substantial correlation between these molecules following spinal nerve injury. It is also interesting to explore how spinal nectin-1 can facilitate a central pain sensation following peripheral nerve injury.

One limitation of the present study is that we did not evaluate the potential role of pentraxin-3/nectin-1 cascades in neuropathic pain in female animals. The first reason is the resource constraints of funding and the pressure for minimal animal usage. Second, all experiments were performed on males to avoid the distraction of fluctuations in the sex hormone in females. Third, recent literature has demonstrated that neuronal modifications and synaptic plasticity in the pathogenesis of pathological pain and central pain sensitization may be sex-independent [29,30]. Actually, mechanical allodynia caused by peripheral nerve trauma is effectively alleviated following intrathecal administration of an NMDA receptor antagonist and CXCR2 (primarily expressed in neurons) antagonist in both male and female animals [29,30]. Thus, to translate the interesting discovery to females, whether there is a sex difference in pentraxin-3/nectin-1 cascades in nociceptive synaptic plasticity after nerve trauma should be taken into consideration in future. Another weakness is that we did not evaluate whether the inhibition of pentraxin-3/nectin-1 cascades impaired the existing neuropathic pain, which needs further investigation. Moreover, further studies are required to examine whether the pentraxin-3 and nectin-1 cascade in the spinal dorsal horn is implicated in other pain syndromes, such as inflammatory pain, bone cancer pain, chemotherapy-induced neuropathy and opioid-induced hyperalgesia.

## 5. Conclusions

Overall, the presented data reveal a novel mechanism by which abnormal nectin-1 expression induced by pentraxin-3 contributes to spinal nerve damage-related peripheral neuropathic pain in male mice. Pharmacological intervention with pentraxin-3/nectin-1 protects against the establishment of pain behaviors. The latter hypothesis can be extended to other nociceptive disorders related to the pentraxin-3 cascade and may help develop novel treatment approaches for a more targeted neurotherapy of pathologic pain.

## Figures and Tables

**Figure 1 brainsci-12-00648-f001:**
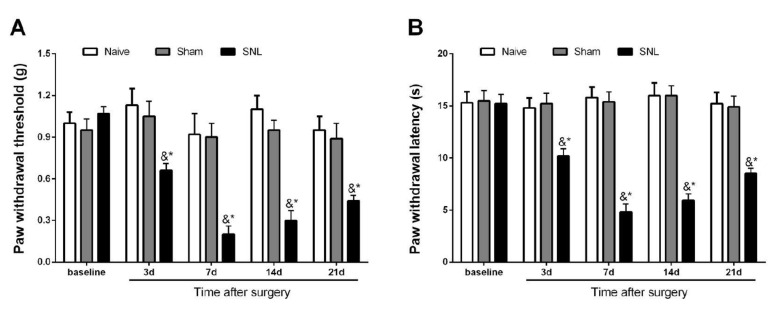
Mechanical allodynia and thermal hyperalgesia following spinal nerve injury. (**A**) Initiation and maintenance of mechanical allodynia as measured by paw withdrawal (PW) threshold in the von Frey test following spinal nerve ligation (SNL). (**B**) Initiation and maintenance of heat hyperalgesia as evaluated by PW latency in the hot plate test following SNL. The data are expressed as the mean ± SEM (n = 6) and analyzed by two-way ANOVA with Bonferroni post hoc comparisons. & *p* < 0.05 vs. baseline, * *p* < 0.05 vs. group Sham.

**Figure 2 brainsci-12-00648-f002:**
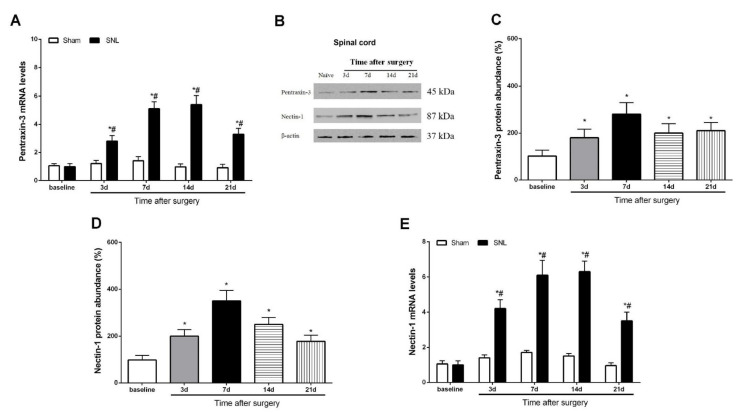
Time course of pentraxin-3 and nectin-1 expression in the spinal dorsal horn following peripheral nerve trauma. The spinal dorsal horn L_3–5_ segments were collected for RT-qPCR and Western blot analyses. The values of the pentraxin-3 (**A**) mRNA expression levels were presented as fold increase over baseline and normalized according to the expression of GAPDH. (**B**–**D**) Western blot analysis was used to detect the spinal levels of pentraxin-3 and nectin-1 protein following spinal nerve ligation (SNL). The spinal expression levels of nectin-1 (**E**) mRNA were presented as fold increase over baseline levels and normalized according to the expression of glyceraldehyde 3-phosphate dehydrogenase (GAPDH). The data were expressed as the mean ± SEM (n = 6) and analyzed by one-way ANOVA with Bonferroni post hoc comparisons. * *p* < 0.05 vs. baseline, # *p* < 0.05 vs. group Sham.

**Figure 3 brainsci-12-00648-f003:**
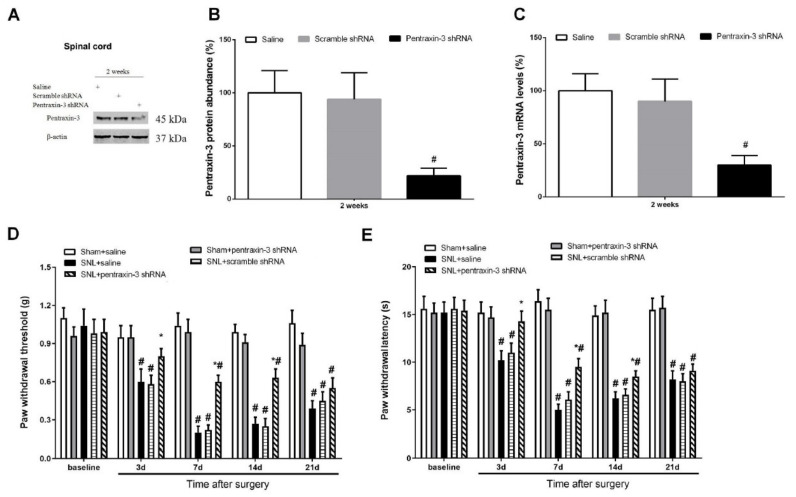
Spinal pentraxin-3 knockdown impairs neuropathic pain associated with peripheral nerve trauma. (**A**,**B**) Western blot analysis demonstrated that the spinal expression of pentraxin-3 protein was downregulated at 2 weeks following intrathecal injection of pentraxin-3 shRNA. The expression levels of pentraxin-3 (**C**) mRNA were presented as fold increase over group Saline and normalized according to the expression levels of glyceraldehyde 3-phosphate dehydrogenase (GAPDH). The results are indicative of the mean ± SEM (n = 6) and analyzed by one-way ANOVA with Bonferroni post hoc comparisons. # *p* < 0.05 vs. group Saline. (**D**,**E**) Spinal pentraxin-3 deficiency prevented mechanical allodynia and heat hyper-nociception due to spinal nerve ligation (SNL). The results are indicative of the mean ± SEM (n = 6) and analyzed by two-way ANOVA with Bonferroni post hoc comparisons. # *p* < 0.05 vs. group Sham + saline, * *p* < 0.05 vs. group SNL + saline.

**Figure 4 brainsci-12-00648-f004:**
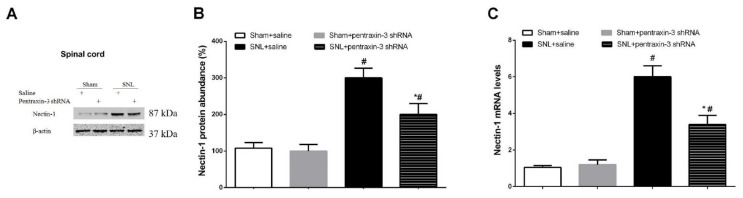
Spinal pentraxin-3 knockdown reduces spinal nectin-1 expression following peripheral nerve injury. (**A**,**B**) Western blot analysis was used for the detection of the spinal levels of nectin-1 protein following pentraxin-3 knockdown in SNL mice. The expression levels of nectin-1 (**C**) mRNA were presented as fold increase over group Sham+saline and normalized according to the expression levels of glyceraldehyde 3-phosphate dehydrogenase (GAPDH). The results are indicative of the mean ± SEM (n = 6) and analyzed by one-way ANOVA with Bonferroni post hoc comparisons. # *p* < 0.05 vs. group Sham + saline, * *p* < 0.05 vs. group SNL + saline.

**Figure 5 brainsci-12-00648-f005:**
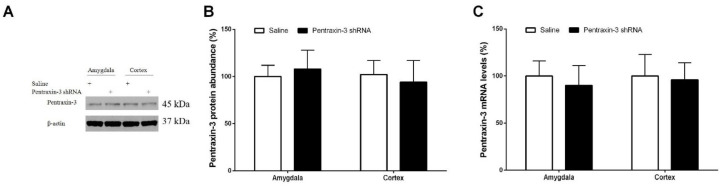
Intrathecal injection of pentraxin-3-shRNA does not impair the expression of pentraxin-3 in the amygdala and cortex. (**A**,**B**) Western blot analysis demonstrated the expression of pentraxin-3 protein in the amygdala and cortex at 2 weeks following intrathecal administration of pentraxin-3 shRNA and normal saline. The expression levels of pentraxin-3 (**C**) mRNA were presented as fold increase over group Saline and normalized according to the expression levels of glyceraldehyde 3-phosphate dehydrogenase (GAPDH). The results are indicative of the mean ± SEM (n = 6) and analyzed by one-way ANOVA with Bonferroni post hoc comparisons.

**Figure 6 brainsci-12-00648-f006:**
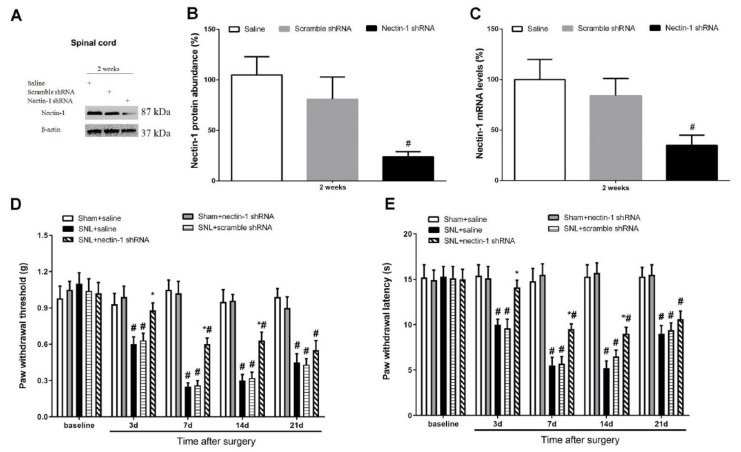
Spinal nectin-1 knockdown impairs neuropathic pain associated with peripheral nerve trauma. (**A**,**B**) Western blot analysis indicated that the spinal expression of nectin-1 protein was downregulated at 2 weeks following intrathecal delivery of nectin-1 shRNA. The expression levels of nectin-1 (**C**) mRNA were presented as fold increase over group Saline and normalized according to the expression levels of glyceraldehyde 3-phosphate dehydrogenase (GAPDH). The results are indicative of the mean ± SEM (n = 6) and analyzed by one-way ANOVA with Bonferroni post hoc comparisons. # *p* < 0.05 vs. saline group. (**D**,**E**) Spinal nectin-1 deficiency prevented mechanical allodynia and heat hyper-nociception due to spinal nerve ligation (SNL). The results are indicative of the mean ± SEM (n = 6) and analyzed by two-way ANOVA with Bonferroni post hoc comparisons. # *p* < 0.05 vs. Sham + saline group, * *p* < 0.05 vs. SNL + saline group.

**Figure 7 brainsci-12-00648-f007:**
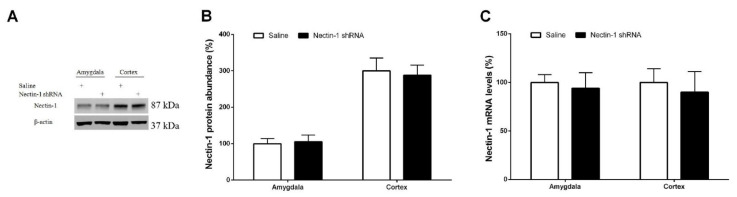
Intrathecal injection of nectin-1-shRNA does not impair the expression of nectin-1 in the amygdala and cortex. (**A**,**B**) Western blot analysis demonstrated the expression of nectin-1 protein in the amygdala and cortex at 2 weeks following intrathecal administration of nectin-1 shRNA and normal saline. The expression levels of nectin-1 (**C**) mRNA were presented as fold increase over group Saline and normalized according to the expression levels of glyceraldehyde 3-phosphate dehydrogenase (GAPDH). The results are indicative of the mean ± SEM (n = 6) and analyzed by one-way ANOVA with Bonferroni post hoc comparisons.

**Figure 8 brainsci-12-00648-f008:**
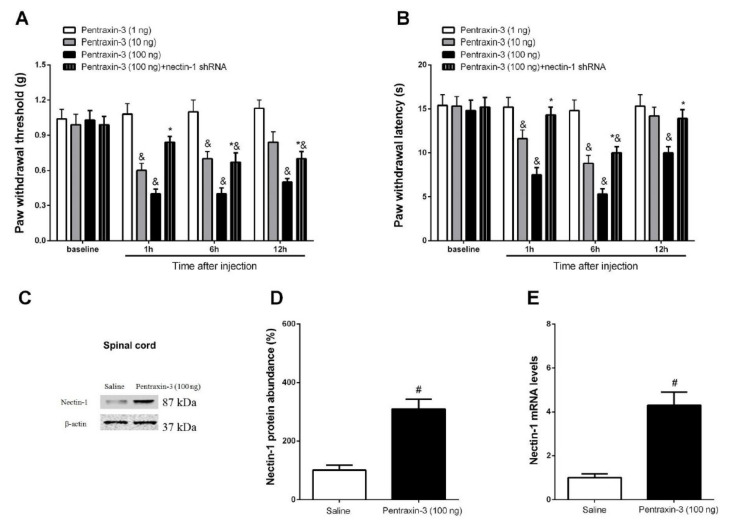
Acute pain is elicited by intrathecal injection of pentraxin-3 and relieved by spinal knockdown of nectin-1. Naïve mice or nectin-1 shRNA treated mice received the intrathecal injection with exogenous pentraxin-3 protein (on day 14 following gene knockdown). The paw withdrawal (PW) threshold (**A**) and PW latency (**B**) of the left hind paw were recorded following pentraxin-3 exposure. The results are indicative of the mean ± SEM (n = 6) and analyzed by two-way ANOVA with Bonferroni post hoc comparisons. The spinal dorsal horn L_3–5_ segments were collected at 6 h following pentraxin-3 treatment for RT-qPCR and Western blot analyses. (**C**,**D**) Western blot analysis indicated the elevation in spinal expression of nectin-1 protein following acute exposure to recombinant pentraxin-3. The mRNA expression levels of nectin-1 (**E**) were presented as fold increase over the saline group and normalized according to the expression levels of glyceraldehyde 3-phosphate dehydrogenase (GAPDH). The results are indicative of the mean ± SEM (n = 6) and analyzed by one-way ANOVA with Bonferroni post hoc comparisons. & *p* < 0.05 vs. baseline, * *p* < 0.05 vs. group Pentraxin-3 (100 ng), # *p* < 0.05 vs. group Saline.

## Data Availability

All data relevant to the research are included in the paper for figures. Data are available from the corresponding author upon reasonable request.

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
