# Peer review of "Pentraxin-3 in the Spinal Dorsal Horn Upregulates Nectin-1 Expression in Neuropathic Pain after Spinal Nerve Damage in Male Mice"

_brainsci, 2022, doi:10.3390/brainsci12050648_

Round 1
Reviewer 1 Report
Dear Authors,
This article is on evaluating the expression of pentraxin-3 in the spinal dorsal horn and its impact on the upregulation of nectin-1 in nociceptive sensitization. The study is well designed, and methodology is properly chosen to address the objectives. Introduction and the background of the condition and the related literature are discussed elaborately.
Suggestions:
The methodology can be expanded, and more details can be added though it points to the previously established protocols.
The full manuscript needs to be read thoroughly to check whether the results are conveyed correctly as per the real data, as in most of the results, the statistical significance mentioned in the results section does not correlate with the graphs. This questions the reliability of the experiments conducted or the whole manuscript itself.
mRNA levels were evaluated using PCR and protein levels with Western blots. Gene expression was used to mention mRNA levels and relative expression level % was used for western blot results. This is confusing as gene expression means the translated protein and not the mRNA. Hence, it is suggested to use ‘mRNA levels’ for PCR result and ‘protein abundance’ for western blot results.
The methodology section mentions using three different tissues such as the spinal dorsal horn, cortex and the amygdala for both the PCR and the western blot experiments. However, data or the results does not indicate of mentions the tissues used. Please mention the specific in the result section and also in the figures.
Is there any specific reason for using SEM and not standard deviation??..Using SD would be more appropriate.
Fig1: Results section states p<0.001 which is not reflected in the histogram and the fig legend.
Fig2:
Gene expression of the pentraxin3 and nectin1 can be labelled as mRNA and western blot results or the histograms with relative expression % as protein abundance. This is in case if I have understood the in the right way. Otherwise, rephrase it accordingly.
Statistical significance is p<0.05 as per the histogram (relative expression %), however the results state it as p< 0.001 which is highly significant. Visually, the histogram gives an overall idea of having p<0.05. So, either the statement or the histogram is wrong, and the data is not an actual representation of the experiment conducted. Only the mRNA levels look like p<0.001 significance.
Also, the western blots of the sham control are not included in the figures. It would be great if sham is added to the manuscript if the data is available.
Fig3:
Similar changes in terms of mRNA level and protein abundance can be used here too.
Pentraxin is knocked down in the mice by administering shRNA and it is not a therapy as such, so it can be directly mentioned as gene knockdown and not as gene therapy.
Also, the discrepancy in the statistical significance has to be fixed.
Fig4:
Please fix the term mRNA and protein abundance. Follow as previously mentioned.
Fig6:
Please change the color or the pattern of the SNL+scramble+shRNA and SNL+nectin-1 shRNA
Check the real p values and the significance of the data presented
Fig8: rephrase the terms nectin 1 relative expression % and gene expression
Check the real p values and the significance of the data presented
In discussion section of the manuscript, Caspase cascade and its blockade are mentioned in relation to the AMPA induced pain. Though the Caspases are important it does not add up to the content of this manuscript. If possible, the mRNA and protein levels are checked in the tissues used in this experiment or delete this paragraph from the manuscript to keep it simple.
Author Response
Dear Editors and reviewers,
Thank you very much for your careful review and constructive suggestions with regard to our manuscript “Pentraxin-3 in the spinal dorsal horn upregulates nectin-1 expression in neuropathic pain after spinal nerve damage in male mice
” (ID: brainsci-1693018). Those comments are helpful for authors to revise and improve our paper. We have studied comments carefully and tried our best to revise and improve the manuscript and made great changes in the manuscript according to the reviewers’ good comments. Revised portion is marked in red in the paper. We appreciate for Editors/Reviewers’ warm work earnestly, and hope that the corrections will meet with approval. Please feel free to contact us with any questions and we are looking forward to your consideration. The main corrections in the paper and the responds to the reviewer’s and editorial comments are as following.
Responds to the reviewers’ comments:
Reviewer #1:
- Response to comment: The methodology can be expanded, and more details can be added though it points to the previously established protocols.
Responds: I quite appreciate your insightful consideration and comment. Considering your suggestion, we have revised and provided a more detailed description in Methods section. We hope that the revision is more acceptable and reasonable.
- Response to comment: The full manuscript needs to be read thoroughly to check whether the results are conveyed correctly as per the real data, as in most of the results, the statistical significance mentioned in the results section does not correlate with the graphs. This questions the reliability of the experiments conducted or the whole manuscript itself.
Responds: I quite appreciate your insightful consideration and comment. We are so sorry for our negligence and inappropriate description. We have revised and re-written the results section after we checked all statistical significance. We hope that the revision is more acceptable and reasonable.
- Response to comment: mRNA levels were evaluated using PCR and protein levels with Western blots. Gene expression was used to mention mRNA levels and relative expression level % was used for western blot results. This is confusing as gene expression means the translated protein and not the mRNA. Hence, it is suggested to use ‘mRNA levels’ for PCR result and ‘protein abundance’ for western blot results.
Responds: I quite appreciate your insightful consideration and comment. We are so sorry for our negligence and inappropriate description. We have revised and re-written all the figures according to your suggestions. We hope that the revision is more acceptable and reasonable.
- Response to comment: The methodology section mentions using three different tissues such as the spinal dorsal horn, cortex and the amygdala for both the PCR and the western blot experiments. However, data or the results does not indicate of mentions the tissues used. Please mention the specific in the result section and also in the figures.
Responds: I quite appreciate your insightful consideration and comment. We are so sorry for our negligence and insufficient description. We have revised and re-written all the figures, results and figure legends marked in red according to your suggestions. We hope that the revision is more acceptable and reasonable.
- Response to comment: Is there any specific reason for using SEM and not standard deviation? Using SD would be more appropriate.
Responds: I quite appreciate your insightful consideration and comment. Considering your suggestion, we are willing to interpret it. The reason that data were expressed as mean ± SEM was based on previous numerous animal reports for comparable nociceptive behavioral experiments and biochemical assays in pain research (Nature. 2021; 591(7849):275-280; Nat Commun. 2021; 12(1):4558; J Clin Invest. 2020; 130(7):3603-3620; Neuron. 2021; 109(17):2691-2706.e5; J Neurosci 2019; 39(35): 6848-6864; Brain Behav Immun 2018; 72: 34-44; Nat Neurosci 2017; 20(7):917-26; Nat Commun 2019; 10(1):5643; J Clin Invest 2014; 124(3):1173–86; J Clin Invest 2018; 128(8):3568-82; Cell Rep 2018; 25(1):168-82; Neuron 2016; 92:1279-93; Proc Natl Acad Sci USA 2016; 113:E6686–95; PAIN 2021; 162(1): 124-134). We hope that SEM is also acceptable and reasonable in our pain research in mice.
- Response to comment: Fig1: Results section states p<0.001 which is not reflected in the histogram and the fig legend.
Responds: I quite appreciate your insightful consideration and comment. We are so sorry for our negligence and insufficient description. We have revised and re-written results section after we checked all statistical significance. We hope that the revision is more acceptable and reasonable.
- Response to comment: Fig2: Gene expression of the pentraxin3 and nectin1 can be labelled as mRNA and western blot results or the histograms with relative expression % as protein abundance. This is in case if I have understood the in the right way. Otherwise, rephrase it accordingly.
Responds: I quite appreciate your insightful consideration and comment. We are so sorry for our negligence and inappropriate description. We have revised and re-written all the figures according to your suggestions. We hope that the revision is more acceptable and reasonable.
- Response to comment: Fig 2: Statistical significance is p<0.05 as per the histogram (relative expression %), however the results state it as p< 0.001 which is highly significant. Visually, the histogram gives an overall idea of having p<0.05. So, either the statement or the histogram is wrong, and the data is not an actual representation of the experiment conducted. Only the mRNA levels look like p<0.001 significance.
Responds: I quite appreciate your insightful consideration and comment. We are so sorry for our negligence and insufficient description. We have revised and re-written results section after we checked all statistical significance. We hope that the revision is more acceptable and reasonable.
- Response to comment: Fig 2: Also, the western blots of the sham control are not included in the figures. It would be great if sham is added to the manuscript if the data is available.
Responds: I quite appreciate your insightful consideration and comment. Considering your suggestion, we are willing to interpret it. First, we did not detect any differences of nociceptive behaviors between naïve and sham animals. Second, we did not observe any differences of pentraxin-3/nectin-1 mRNA levels between naïve and sham animals. So we supposed that there is no difference of pentraxin-3/nectin-1 protein abundance between naïve and sham animals. That is why we did not evaluate the protein levels of sham group in Western blots in figure 2. Sure, I agree with you. It would be great if this data was included in paper. This is one potential limitation in our present study.
- Response to comment: Fig 3: Similar changes in terms of mRNA level and protein abundance can be used here too.
Responds: I quite appreciate your insightful consideration and comment. We are so sorry for our negligence and insufficient description. We have revised and re-written all figures. We hope that the revision is more acceptable and reasonable.
- Response to comment: Fig 3: Pentraxin is knocked down in the mice by administering shRNA and it is not a therapy as such, so it can be directly mentioned as gene knockdown and not as gene therapy.
Responds: I quite appreciate your insightful consideration and comment. We are so sorry for our negligence and inappropriate description. We have revised and re-written results section and figure legends. We hope that the revision is more acceptable and reasonable.
- Response to comment: Fig 3: Also, the discrepancy in the statistical significance has to be fixed.
Responds: I quite appreciate your insightful consideration and comment. We are so sorry for our negligence and insufficient description. We have revised and re-written results section after we checked all statistical significance. We hope that the revision is more acceptable and reasonable.
- Response to comment: Fig 4: Please fix the term mRNA and protein abundance. Follow as previously mentioned.
Responds: I quite appreciate your insightful consideration and comment. We are so sorry for our negligence and inappropriate description. We have revised and re-written all figures. We hope that the revision is more acceptable and reasonable.
- Response to comment: Fig 6: Please change the color or the pattern of the SNL+scramble+shRNA and SNL+nectin-1 shRNA.
Responds: I quite appreciate your insightful consideration and comment. We are so sorry for our negligence and inappropriate pattern in figures. We have revised and re-written figure 3 and figure 6 according to your suggestions. We hope that the revision is more acceptable and reasonable.
- Response to comment: Fig 6: Check the real p values and the significance of the data presented.
Responds: I quite appreciate your insightful consideration and comment. We are so sorry for our negligence and insufficient description. We have revised and re-written results section after we checked all statistical significance. We hope that the revision is more acceptable and reasonable.
- Response to comment: Fig 8: rephrase the terms nectin 1 relative expression % and gene expression.
Responds: I quite appreciate your insightful consideration and comment. We are so sorry for our negligence and inappropriate description. We have revised and re-written all figures. We hope that the revision is more acceptable and reasonable.
- Response to comment: Fig 8: Check the real p values and the significance of the data presented.
Responds: I quite appreciate your insightful consideration and comment. We are so sorry for our negligence and insufficient description. We have revised and re-written results section after we checked all statistical significance. We hope that the revision is more acceptable and reasonable.
- Response to comment: In discussion section of the manuscript, Caspase cascade and its blockade are mentioned in relation to the AMPA induced pain. Though the Caspases are important it does not add up to the content of this manuscript. If possible, the mRNA and protein levels are checked in the tissues used in this experiment or delete this paragraph from the manuscript to keep it simple.
Responds: I quite appreciate your insightful consideration and comment. We are so sorry for our inappropriate discussion. We have revised and deleted this part according to your suggestion. We hope that the revision is more acceptable and reasonable.
Special thanks to you for your good comments.
In all, we found that the reviewers’ and editorial comments are quite helpful, and we revised our paper point-by-point and tried to avoid any error and negligence. Simultaneously, we have asked several colleagues who are skilled authors of English language papers to check the manuscript. Once again, thank you for your careful review and constructive suggestions!
The manuscript has been re-submitted to your journal. We are convinced that several aspects of this manuscript will make it interesting to general readers of Brain Sciences.
Thank you very much for your kind attention and consideration. Any suggestions and criticism for the consideration of possible publication of the manuscript will be appreciated. I am looking forward to hearing from you.
Sincerely yours,
Wenli Yu, Ph.D
Department of Anesthesiology
Tianjin First Center Hospital, Tianjin 300192, China
Reviewer 2 Report
Here, Zhu and his colleagues assessed spinal pentraxin-3/nectin-1 expression in SNL mice, as well as the relevant behavior. Their results demonstrate that pentraxin-3/nectin-1 is required for SNL-triggered neuropathic pain. This is a straightforward work, showing convinced behavior, PCR and WB results. Pentraxin-3/nectin-1 is well-studied, hot topic in neuroscience/cancer field, and this study show us a promising role of Pentraxin-3/nectin-1 in neuropathic pain. These interesting and informative outcome will provide readers a promising role of pentraxin-3/nectin-1 as a potential target in pain treatment.
However, there are some items that need further clarification. The manuscript may benefit from the following recommendations below.
- The authors only use male information mice in this studies, please classify this point.
- The authors only use 2 hours for habituation prior to behavior test, is this sufficient, please classify this point.
- Majority of the PCR and WB figures needs re-format. The bands are not clear, and the molecular weight are not mentioned. Authors should especially pay attention on the legend of y-axis in PCR and WB.
- The author shows the preventing role of Pentraxin-3/nectin-1 shRNA on neuropathic pain, it will be perfect they also speculate whether it can also reverse the pain behavior on the existing pain.
- In discussion section, authors may explain the relationship between pentraxin-3 and nectin-1, this might be useful to the readers.
Author Response
Dear Editors and reviewers,
Thank you very much for your careful review and constructive suggestions with regard to our manuscript “Pentraxin-3 in the spinal dorsal horn upregulates nectin-1 expression in neuropathic pain after spinal nerve damage in male mice
” (ID: brainsci-1693018). Those comments are helpful for authors to revise and improve our paper. We have studied comments carefully and tried our best to revise and improve the manuscript and made great changes in the manuscript according to the reviewers’ good comments. Revised portion is marked in red in the paper. We appreciate for Editors/Reviewers’ warm work earnestly, and hope that the corrections will meet with approval. Please feel free to contact us with any questions and we are looking forward to your consideration. The main corrections in the paper and the responds to the reviewer’s and editorial comments are as following.
Responds to the reviewers’ comments:
Reviewer #2:
- Response to comment:The authors only use male information mice in this study, please classify this point.
Responds: I quite appreciate your insightful consideration and comment. Considering your suggestion, we are willing to interpret it. We did take only male mice in this study, just as most of basic pain research. The first reason is that the pressure for applicability is at odds with the resource constraints of funding and the need to reduce animal usage. Second, we focused on males to avoid confounding sex-hormone fluctuations in females. Third, according to related literature (Neurosci Bull 2018, 34(1):98-108; Nat Neurosci 2015, 18: 1081-3), we considered that neuronal regulation and synaptic plasticity in pain mechanism may be sex-independent. Actually, sex dimorphism in microglial modulation of pathologic pain has been emphasized (Neurosci Bull 2018, 34(1):98-108; Anesthesiology 2018, 129(2):343-66). Spinal Toll-like receptor 4, a key receptor for microglia activation, involves the development of inflammatory pain and neuropathic pain exclusively in male mice (J Neurosci 2011; 31(43):15450-4). Nerve injury-induced mechanical allodynia is only alleviated in male mice after spinal administration of microglia inhibitor minocycline, microglia toxin, p38 inhibitor, or P2X4 blocker (Neurosci Bull 2018, 34(1):98-108). Conversely, mechanical hypersenstivity after nerve injury is attenuated in both males and females equally after intrathecal delivery of astrocyte toxin L-AA and central inhibition of JNK and ERK, two MAP kinases important for astrocytic signaling (Neurosci Bull 2018, 34(1):98-108). Consistently, intrathcal injection of NMDAR antagonist and CXCR2 (primarily expressed in neurons) antagonist reduce nerve injury-caused mechanical pain behaviors in both sexes (Neurosci Bull 2018, 34(1):98-108; Nat Neurosci 2015, 18: 1081-3). Given that spinal astrocytic and neuronal signaling in synaptic transmission and central sensitization appear to be sex-independent, we did not investigate sex difference in pentraxin-3/nectin-1 pathway in synaptic plasticity and pain transduction in our model. Certainly, to translate these findings to females, we should take this into consideration in future. We have briefly added this comment in discussion section in Page 19 marked in red. Also, we have clarified that we used male animals throughout the manuscript.
- Response to comment:The authors only use 2 hours for habituation prior to behavior test, is this sufficient, please classify this point.
Responds: I quite appreciate your insightful consideration and comment. Considering your suggestion, we are willing to interpret it. Actually, all male animals were acclimated to the testing environment for 3 days prior to any nociceptive behavior examinations. To assess mechanical sensitivity, mice were placed in a Plexiglas box and allowed 2 hours for further acclimation prior to the study. We are so sorry for our insufficient and inappropriate methods description. Considering your useful suggestion, we have briefly added related method information and descriptions in methods section in Page 8 marked in red.
- Response to comment:Majority of the PCR and WB figures needs re-format. The bands are not clear, and the molecular weight are not mentioned. Authors should especially pay attention on the legend of y-axis in PCR and WB.
Responds: I quite appreciate your insightful consideration and comment. Considering your suggestion, we have revised and re-made all figures. Also, the molecular weights are included in the figures. We hope that the revision is more acceptable.
- Response to comment:The author shows the preventing role of Pentraxin-3/nectin-1 shRNA on neuropathic pain, it will be perfect they also speculate whether it can also reverse the pain behavior on the existing pain.
Responds: I quite appreciate your insightful consideration and comment. Considering your suggestion, we are willing to interpret it. Actually, we really wanted to test whether pentraxin-3/nectin-1 could reverse the existing pain, but we did not find any effective inhibitors on pentraxin-3/nectin-1. It does take 2 weeks to achieve stable reduction in the pentraxin-3/nectin-1 proteins after intrathecal injection of shRNA, so shRNA is hardly used to knockdown pentraxin-3/nectin-1 proteins in mice that have underwent nerve trauma. It is difficult to choose the right time to inject shRNA for testing its role in the existing pain. Anyway, I agree you that this is one potential limitation. We have briefly added this limitation in Discussion section in Page 20 marked in red.
- Response to comment:In discussion section, authors may explain the relationship between pentraxin-3 and nectin-1, this might be useful to the readers.
Responds: I quite appreciate your insightful consideration and comment. Considering your suggestion, we are willing to interpret it. pentraxin-3 has been shown to promote the formation and development of functional synapses by increasing AMPA receptor trafficking in the rodent neuronal synapses. Nectin-1, which belongs to immunoglobulin-like cell adhesion molecules, is localized in excitatory glutamatergic synapses and constitutes an important determinant of synapse generation and function. Nonetheless, it is virtually unknown regarding the interaction of spinal pentraxin-3 and nectin-1 in neuropathic pain initiation and sustenance. Our current study provided multiple data for supporting spinal nectin-1, which is the downstream target of pentraxin-3, is one of the most critical factors contributing to the mechanism of SNL-associated neuropathic pain. Initially, SNL intervention persistently upregulated nectin-1 in the spinal dorsal horn. Secondly, knockdown of spinal nectin-1 successfully reduced postoperative pain upon SNL surgery. Thirdly, following the inhibition of central pentraxin-3, the reversal of the effects associated with nectin-1 overexpression following spinal nerve injury was achieved. Fourthly, exogenous pentraxin-3 after intrathecal administration induced transient pain and spinal nectin-1 overload, which was reversed by co-application of nectin-1 shRNA. These results elucidated that spinal pentraxin-3 increases the expression levels of nectin-1, triggering SNL-related pro-nociception perception in male animals. We have included these rationale and results description in Introduction section in Page 5 and in Discussion section in Page 18-19 marked in red. We hope that the revision is more reasonable and acceptable.
Special thanks to you for your good comments.
In all, we found that the reviewers’ and editorial comments are quite helpful, and we revised our paper point-by-point and tried to avoid any error and negligence. Simultaneously, we have asked several colleagues who are skilled authors of English language papers to check the manuscript. Once again, thank you for your careful review and constructive suggestions!
The manuscript has been re-submitted to your journal. We are convinced that several aspects of this manuscript will make it interesting to general readers of Brain Sciences.
Thank you very much for your kind attention and consideration. Any suggestions and criticism for the consideration of possible publication of the manuscript will be appreciated. I am looking forward to hearing from you.
Sincerely yours,
Wenli Yu, Ph.D
Department of Anesthesiology
Tianjin First Center Hospital, Tianjin 300192, China
Round 2
Reviewer 2 Report
The authors have addressed my questions point-by-point and the MS has improved quite a lot. I have no further concerns.
Thanks